# The Effects of Spinal Manipulation on Oculomotor Control in Children with Attention Deficit Hyperactivity Disorder: A Pilot and Feasibility Study [note 1]

**DOI:** 10.3390/brainsci11081047

**Published:** 2021-08-06

**Authors:** Alice Cade, Kelly Jones, Kelly Holt, Abdul Moiz Penkar, Heidi Haavik

**Affiliations:** 1Centre for Chiropractic Research, New Zealand College of Chiropractic, Auckland 1060, New Zealand; kelly.holt@nzchiro.co.nz (K.H.); amoizp@yahoo.com (A.M.P.); heidi.haavik@nzchiro.co.nz (H.H.); 2National Institute for Stroke and Applied Neurosciences, Auckland University of Technology, Auckland 1142, New Zealand; kejones@aut.co.nz

**Keywords:** attention deficit hyperactivity disorder, spinal manipulation, chiropractic, eye movements, children, randomized controlled pilot trial

## Abstract

Attention Deficit Hyperactivity Disorder (ADHD) is a prevalent, chronic neurodevelopmental disorder that affects oculomotor (eye movement) control. Dysfunctional oculomotor control may result in reading or educational difficulties. This randomized controlled crossover study sought to investigate the feasibility of a larger scale trial and effects of a single session of spinal manipulation on oculomotor control in children with ADHD. Thirty children participated in the study and were randomized into either control-first or spinal manipulation first groups. The results indicate that the trial was feasible. Secondary outcomes showed that there was a significant decrease in reading time after the spinal manipulation intervention compared to the control intervention. Future studies of the effects of spinal manipulation on oculomotor control in children with ADHD are suggested.

## 1. Introduction

The worldwide prevalence of Attention Deficit Hyperactivity Disorder (ADHD) is estimated to be around 5.9–7.1% of the population [1,2], making it the most common neurobehavioral disorder of childhood [3]. ADHD is defined as a chronic neurodevelopmental disorder, characterized by inattention and/or hyperactivity-impulsivity reaching past normal developmental stages [4,5]. Children with ADHD are 7.1 times more likely to be expelled from school and 3.1 times more likely to repeat a grade compared to children without ADHD [6]. Those with ADHD are 4 to 5 times more likely to use special educational services than children without ADHD [7]. Children with ADHD score significantly lower on reading and arithmetic achievement tests than controls [7]. Thirty-nine percent of children with ADHD also experience concurrent reading disabilities [8] and previous research suggests these educational issues may result, in part, from oculomotor (eye movement) dysfunction associated with ADHD [9].

Oculomotor control changes in children with ADHD can lead to a compromised ability to suppress unwanted saccades (short jumps of eye movement off the intended target and not under conscious control [10] and reduced control of steady fixation [11]. These alterations in oculomotor control stem from poor processing speed, sensory filtering, working memory, sensorimotor integration, and executive function difficulties likely related to dysfunction in the prefrontal cortex [12,13]. Oculomotor difficulties adversely affect children’s capacity for cognitive processing of visual information and can make simple reading tasks challenging [14]. Specifically, in children with ADHD, oculomotor deficits may present clinically as longer fixations, more regressive saccades, and slower, less accurate word recognition [11]. Together, these visual difficulties make reading challenging for children, more time consuming, and render it more difficult to comprehend read materials [9,15].

Oculomotor control changes have long been associated with spinal dysfunction with the causative factor likely being related to disorganized neck proprioceptive (defined as aberrant sensory information from cervical mechanoreceptors [16]) activity [17,18]. Palmgren et al., (2006) [19] have shown that spinal manipulation can improve neck proprioception, as measured by improved sensorimotor integration and improved eye-tracking control. Spinal manipulation alters sensorimotor integration [20] and specific areas of the brain important for accurate oculomotor control [21]. Thus, it is possible that spinal manipulation may help improve oculomotor function for children with ADHD. This, in turn, may assist children with ADHD in the ease in which they can read material.

This study aimed to determine the feasibility of conducting a randomized controlled trial to assess the effects of spinal manipulation on oculomotor control in children with ADHD. This study also sought to investigate the feasibility of providing a spinal manipulation to children with ADHD in a study situation and to assess its preliminary efficacy for improving oculomotor and reading outcomes.

## 2. Materials and Methods

### 2.1. Design

This was a randomized controlled pilot study that used a cross over design [22] with a one-week minimum washout period. Ethical approval for this study was obtained from the Health and Disability Ethics Committee (HDEC 16/NTA/56) and the Auckland University of Technology Ethics Committee (AUTEC 16/311). The trial was registered with the Australian New Zealand Clinical Trials Registry (ANZCTR) ACTRN12616001448437.

### 2.2. Participants

Eligible participants were children aged 8–15 years who had a parent-reported diagnosis of ADHD. Children medically diagnosed with co-morbid conditions that seriously altered their reading ability (major visual disabilities or reading deficits from sources other than ADHD) were excluded. Children were recruited from participating support groups, chiropractic practices, and Facebook advertising.

### 2.3. Procedures

Following screening and consent, participants were randomized for the order of intervention (chiropractic first or control first), and balanced for age and gender, using a computer-generated block randomization sequence using an online program (QMinim, version 0.3, QMinim Online Minimization, Iran [23]. One parent of each child also participated in the study. Due to the nature of the interventions and to the fact that child participants were required to have a guardian present during either intervention, blinding of participants was not possible. Data analyses were blinded by recording data with an associated participant identification code, then replacing the participant code with an unrelated numerical code before analysis, thereby removing any associated participant identifiers. The preliminary analysis of oculomotor data was also automated and did not require human intervention.

Prior to either intervention, baseline oculomotor data was collected as described below. Children randomized to the intervention arm received a single session of spinal manipulation. Full spine spinal assessment and intervention was carried out as part of the spinal manipulation. The entire spine and sacroiliac joints were assessed for spinal dysfunction and corrected where deemed necessary. The current understanding of spinal dysfunction (also known as vertebral subluxation) is “a self-perpetuating, central segmental motor control problem that involves a joint, such as a vertebral motion segment, that is not moving appropriately, resulting in ongoing maladaptive neural plastic changes that interfere with the central nervous system’s ability to self-regulate, self-organize, adapt, repair, and heal” [24,25]. The clinical indicators that were used to assess the function of the spine included joint tenderness to palpation, restricted inter-segmental range of motion, asymmetric intervertebral muscle tension, and abnormal spinal joint play. All of these are known clinical indicators of spinal dysfunction [26]. Any spinal adjustments carried out were high-velocity, low-amplitude thrusts to the spine or pelvic joints, a standard spinal adjustment technique used by chiropractors [20]. This adjustment technique has also been previously used in studies that have investigated neurophysiological effects of chiropractic intervention chiropractors [20].

Children randomized to the control intervention underwent a series of passive and active movements of the head, spine, and body. No spinal adjustments were performed during any control. This active control was not intended to act as a sham for spinal manipulation but to be a physiological control for any possible changes that may occur due to cutaneous, muscular, or vestibular input from the passive and active movements used in preparing for a spinal adjustment [20]. After each intervention, oculomotor function was again assessed.

At a minimum 7 days after the initial intervention all child participants were reassessed pre and post the alternative intervention. Finally, parents and children were presented with the post study questionnaire and asked to either fill it out at the time or complete it at home and return it to the chief investigator. The post study questionnaire included a participant identification code, a Likert scale grading statements about the study, and a section recording child and parent’ impressions of the study.

### 2.4. Outcome Measures

Trial feasibility was assessed quantitatively and qualitatively by (i) recruitment efficiency; (ii) randomization; and (iii) follow-up completion rates. Intervention feasibility (secondary endpoint) was measured qualitatively by parents regarding their impressions of the intervention.

Preliminary efficacy of the intervention was assessed using several measures of oculomotor control. The Eye Tribe tracker (Eye Tribe, Denmark) is a computerized assessment tool that calculates the location gaze by information extracted from participants’ face and eyes. The eye and gaze coordinates are calculated in relation to a screen (a 21-inch monitor), the participant is looking at and is represented by a pair of (x, y) coordinates [27]. This study utilized two gaze-controlled software applications (one for target acquisition and one for the reading task), generating detailed log files to generate measurement results. Reading and target acquisition tasks have previously been used to assess oculomotor control in children with ADHD and has proven to be a reliable measure in recording oculomotor changes in children with ADHD [11,27,28].

#### 2.4.1. Target Acquisition Task

Using the Eye Tribe Tracker, children were asked to locate a visual target (orange dot) flashed on a computer screen and instructed to ignore a purple distractor dot. Any fixations longer than 300 milliseconds [29] on to the visual target (orange dot) or purple distractor dot were recorded as target acquisition time and number of distractions respectively.

#### 2.4.2. Reading Task

Pre-intervention, children were asked to read a short sentence on a computer screen. Post-intervention the reading task was re-tested using a slightly different set of sentences to avoid participants relying on memory to recognize the sentence. Pre- and post-intervention sentences were similar in length, content, and difficulty [30]. This task measured the number and time of forward and reverse fixations (over 300 ms), the number and length (in pixels) of forward and reverse saccades, and total reading time.

### 2.5. Data Analysis

Feasibility of the intervention was analysed using both quantitative and qualitative descriptive methodology [31]. Trial feasibility was assessed using descriptive statistics (number, unadjusted means, standard deviations, and counts) to evaluate recruitment efficiency, acceptance of randomization, and follow-up completion rates. Any procedural issues that were uncovered during the study were also recorded. Content analysis [32,33] of the post study questionnaire was undertaken and presented as a summary of central themes and (if any) sub themes [34].

Preliminary efficacy data descriptive statistics (unadjusted means, standard deviations, and counts) were used to describe the characteristics of the study sample using SPSS version 21 [35]. Shapiro-Wilk tests [35] revealed that the outcome data collected for all tasks were not normally distributed. Accordingly, all data were subjected to a logarithmic or two-step rank-transformation [36,37] to perform multifactorial repeated measure analysis of variance (ANOVA) to assess for within subject differences. Pre- and post-intervention measures (target acquisition time, number of distractors, reading time, number, time and length of forward and reverse fixations and saccades) for each dependent variable and intervention (intervention versus control) were used as outcome measures when assessing the main effects or interaction effects. An a priori pairwise comparison of the pre and post intervention data was carried out when an interactive effect was significant (set at *p* ≤ 0.05). Confidence intervals were given at a two-sided 95% level. No adjustments to p-values were planned or made due to the use of multiple outcome measures as recommend by Feise (2002) [38] and Perneger (1998) [39]. All analyses were performed on the intention-to-treat principle and all available data were used. No missing data imputation was performed.

## 3. Results

### 3.1. Trial Feasibility

Thirty-four participants were screened for eligibility, with four being ineligible (too young and travel restrictions). The participants were randomized to either receive the control intervention first (*n* = 15) or the spinal manipulation first (*n* = 15). Twenty-seven (90%) participants completed the study (13 children who completed the study received the spinal manipulation first and 14 children received the control intervention first). Three participants (10%) were unable to complete the control intervention session after being unable to complete calibrate with the eye tracker for any outcome measure, however they attended both sessions. Overall, 27 sets of complete or partial data were included in the analysis. The equipment was problematic as the Eye Tribe Tracker had some technical drawbacks. Overall, 12 (40%) participants were unable to complete some parts of the assessment due to difficulties with equipment calibration. Those children with partial data sets were included in the analysis.

Participant recruitment was timely (11 weeks); there were no dropouts, complaints, or other non-compliance issues. Sixty-four families expressed interest in participating in this study. All interested parties contacted the research team via email, preferring this method of communication. Thirty-four people responded after the initial study information was sent to them and agreed to be screened for inclusion.

The majority of participating children were recruited through ADHD support groups (57%), private chiropractic practices (23%), and Facebook (10%). The remaining three (10%) participants were identified through the New Zealand College of Chiropractic student clinic. Two children aged six and seven years did not meet the inclusion criteria due to the lower age limit and two were non-consenting, giving travel time as their reason for declining involvement. The remaining 30 individuals (47%) were unable to be contacted again following the provision of study information. Baseline demographic information and medical history are summarized in Table 1.

### 3.2. Feasibility of the Intervention

Of the participants and parents able to fully participate in this study (those able to calibrate with the equipment), only seven of the post-study evaluation questionnaires were completed for nine (30%) children in total (some participants were siblings, and, in those cases, parents completed one questionnaire for both children). Reasons for not completing the questionnaire were not given by parents. A qualitative descriptive content analysis [40,41,42] of participants’ responses revealed a number of themes running through child and parent responses. Themes suggested from parent responses are shown in Table 2. Overall, responses were positive with 86–100% of participants strongly agreeing with statements. There were no Likert scores below a 6 (on a 7-point Likert scale) given to any of the questions. Respondents (29% (*n* = 2)) recognized the difficulty some participants had with calibration and suggested better equipment would improve their experience. The same number (*n* = 2) found the travel time arduous and a need for more locations from which to perform the study was identified. Of the parents who completed the study, 57% reported improvements in how their child felt after receiving the spinal manipulation. Three respondents (43%) reported nothing that they disliked about the study. Finally, three respondents (43%) reported that the researcher was willing to give information on many aspects of ADHD and involved the child participant in said discussions at a level they could understand and interpret.

### 3.3. Preliminary Efficacy Findings

There were unexpected large group differences in the chiropractic and control pre- intervention measures (see Table 2 and Table 3) possibly due to the order each intervention was given in. The participants receiving the spinal manipulation first seemed to carry changes in their oculomotor outcomes into their control session, suggesting that a one-week wash-out period was not long enough between interventions (see: post-hoc analyses). Raw data is available for the preliminary efficacy findings, please see the Appendix A.

#### 3.3.1. Visual Target Acquisition Time

There was no intervention by time interaction for visual target acquisition time (F (1, 438) = 1.47, *p* = 0.2, partial eta squared = 0.003), suggesting no significant difference in the two interventions (Table 2). There was a large decrease in time to target fixation of 643.37 ms after the spinal manipulation, and increased time to target fixation by 1.34 ms after the control intervention. The lack of statistical significance may be due to the order the intervention was given in or that the washout period may have been too short (see: post-hoc analyses).

#### 3.3.2. Total Reading Time

Total reading time was shortened from 4016 ms (SD = 1894) to 3370 ms (SD = 1508) post the control intervention versus 3548 ms (SD = 1637) to 3440 ms (SD = 1971) post the control intervention. There was a significant interaction between intervention type and time (F (1, 384) = 4.42, *p* = 0.03, partial eta squared = 0.012) (Table 3) suggesting a significant difference in the efficacy of the two interventions.

#### 3.3.3. Forward and Reverse Fixations and Saccades

Descriptive statistics are shown in Table 3. The data for these outcome measures were not normally distributed and as there were no non-parametric tests available [43] no statistical analyses were performed.

### 3.4. Post-Hoc Analyses

Due to the unexpected large group differences in the chiropractic and control pre- intervention measures, post-hoc testing was undertaken to investigate any possible effects of intervention order. Figure 1 and Figure 2 show the outcome measures by which order they were delivered in. For both the number of distractions (in the target task) and total reading time (in the reading task), receiving the spinal manipulation first seemed to alter the baseline recordings obtained in the control intervention session. In both the target acquisition and reading tasks, the participants receiving the spinal manipulation first showed a different pattern of findings to the control-first participants. This may indicate that the wash-out period given was insufficient.

## 4. Discussion

This pilot study was designed with specific feasibility aims, in order to test all aspects of its processes for a possible future full-scale trial. This study suggests that a future randomized clinical trial comparing the efficacy of spinal manipulation with that of an active control intervention on oculomotor function in children with ADHD is feasible. However, findings suggest a need for modifications to some study processes, such as improving the sensitivity of eye tracking equipment and moving to a parallel study design. The recommended modifications to some aspects of this study’s processes will enhance the likelihood of gaining more accurate and appropriate outcomes in future work.

Based on this study, sample size calculations using the R software [44] package pwr [45] suggest a sample size of ~20 based on an expected intraclass correlation coefficient of 0.6 and a 95% confidence interval, which is predefined as the lowest acceptable level of test-retest [46,47,48,49].

The secondary findings of this study were that of statistically significant improvements in the sentence task reading time post the spinal manipulation as compared to post the control intervention. These findings provide some support for the hypothesis that a single session of spinal manipulation may improve short-term or perhaps immediate oculomotor control in children with ADHD. The remaining outcome data, not related to reading time, show trends towards improvements post spinal manipulation, but was not statistically significant. Nearly all of the reading and target outcomes improved post spinal manipulation, indicating better reading ability and decreased distractibility [11,50]. The caveat of this discussion is that, excepting reading time, none of the data were statistically significant, thus it can only be said to show trends or possibilities in the data that warrant further investigation. It is possible that this is due to a type II error in that the sample size of this study was not large enough to show the true effect [51]. It is also possible the baseline differences in the chiropractic first or control first participants affected baseline outcome measures, or that the wash-out period given was insufficient.

To date, the effects of spinal manipulation on the neurological function of children diagnosed with ADHD have not been investigated thoroughly and are limited mostly to case studies and retrospective case reviews [52,53]. Previous studies [20,54,55,56] have shown that spinal manipulation affects sensorimotor integration (within the CNS) likely through altered proprioceptive input from spinal origins, which may go some way to explaining the findings of this study. Afferent input from dysfunctional spinal motion segments has been shown to affect oculomotor function in a variety of populations [57,58]. Manipulation of dysfunctional spinal joints can alter somatosensory filtering of afferent inputs [54,59,60,61]. Multiple previous studies have shown that adjusting the spine leads to changes in somatosensory processing, sensorimotor integration, and motor control of both upper and lower limb muscles [19,20,26,62,63,64]. As oculomotor control relies on accurate sensory processing [65], it is possible that aberrant afferent input from the spine [66] could change somatosensory processing enough to alter oculomotor control.

Another potential mechanism for why spinal manipulation may influence oculomotor control could be the influence spinal manipulation has on the prefrontal cortex function [21]. Lelic et al. (2016) reported alterations in the prefrontal cortex processing with chiropractic adjustments to the spine. The prefrontal cortex is the ‘decision-making’ area of the brain that unconsciously decides where the next saccade moves the gaze to [67] and the prefrontal cortex has been reported to be an area of dysfunction in people with ADHD [12,68]. Prefrontal cortex function also influences reaction times, responses to stimuli, and inhibition of irrelevant distraction, all of which are necessary for reading and comprehension [13,28,69,70,71], and it contains the frontal eye fields, which are essential for the control of eye movements [72]. As this area of the brain is morphologically altered in children with ADHD [69,73,74] it is plausible that adjusting the spine of children with ADHD may alter their prefrontal cortex function, in turn altering oculomotor function.

### Limitations

This study has several limitations, the first being the sample size of 27 participants. While the findings of this study suggest that spinal manipulation may offer the potential to positively influence oculomotor control and reading speed in children with ADHD this must be interpreted with caution due to the small sample size. It is also still unclear whether these findings translate into an improvement in reading ability. Furthermore, this study can only report on immediate changes post-chiropractic but cannot speak to longer term effects. Nevertheless, findings suggest that adjusting the spine may be associated with improvements in oculomotor control relating to children’s reading times.

There were also baseline characteristic differences. At this time, it is unknown what influence an adjustment has on sensorimotor integration post 60 min [75], so it is possible that previous spinal manipulation may have altered the baseline outcome measures collected. Post-hoc analysis of data for the target and reading tasks seemed to show that receiving the spinal manipulation before the control intervention affected control-second outcome measures, i.e., the improvements post the spinal manipulation were still present one week later. This suggests that a parallel study design would be a better choice for future studies or that a much longer wash-out period should be used.

The presence of comorbid disorders could have affected the baseline data for the chiropractic or control first interventions as well. The control intervention first group had more participants with comorbid disorders compared to the chiropractic first group (66.7% and 40.0%, respectively). While disorders comorbid to ADHD (dyslexia, dyspraxia, ASD) share similar characteristics to ADHD, they differ in their aetiology and slightly in their areas of brain dysfunction [75]. However, these disorders do seem to also affect oculomotor function [28,76,77] and some disorders can respond to oculomotor retraining [78,79,80]. Inclusion criteria for the current study were also very open and non-restrictive, but the exclusion criteria, while useful for defining participant age, were not specific enough regarding reading ability. The exclusion criteria were any child who had been medically diagnosed with comorbid conditions that seriously altered their reading ability (major visual disabilities or reading deficits from sources other than ADHD). This did not exclude illiterate children who had no medical explanation for their inability to read, or children who were non-verbal (the reading task required answering a question). A future prudent addendum to eligibility would require participant’s to be verbal and able to read at a minimum level.

Additionally, the attending chiropractor had the freedom to use their judgement on where in the spine to deliver an adjustive thrust to the participant, as would happen in normal practice. These factors, while enhancing the external validity and potential generalizability of the study, did provide a trade-off with internal validity as the chiropractic adjustments differed in vertebral level from participant to participant.

Lastly, none of the outcomes tested were strict measures of cognitive ability [81,82]. Improvements in oculomotor function can only infer improvements in reading and cognitive ability [29]. An additional step in the reading task, for future studies, could be to add a ‘correct/incorrect’ metric to each question asked in the sentence-reading task. This may allow a better inference of cognitive processing regarding the sentence content [83], i.e., did the participants actually read the sentence, and more importantly, did they understand the content.

Nevertheless, findings of the current study indicate promise in investigating the effects of spinal manipulation on oculomotor control and reading behaviour in children with ADHD. The choice of study design limits the interpretation of the data collected in this study. Post-hoc analysis of the data collected suggests that the effects of a single session of spinal manipulation may have lasted longer than the washout period used. In effect, receiving the spinal manipulation first seemed to affect the control values recorded in the second assessment session, suggesting that a parallel group design would have been a better choice for this study.

## 5. Conclusions

This study has shown that with some alterations to study design, as discussed above, that it is feasible to study the effects of spinal manipulation on oculomotor control in children with ADHD. It has provided greater clarification surrounding the study procedures, equipment, and management, likely leading to future research opportunities that will be more reliable and generally applicable to the greater ADHD population. Further, study findings indicate that a single session of spinal manipulation shows promise for altering oculomotor function in children with ADHD, when compared to an active control, however further research is required to explore these findings fully.

## Figures and Tables

**Figure 1 brainsci-11-01047-f001:**
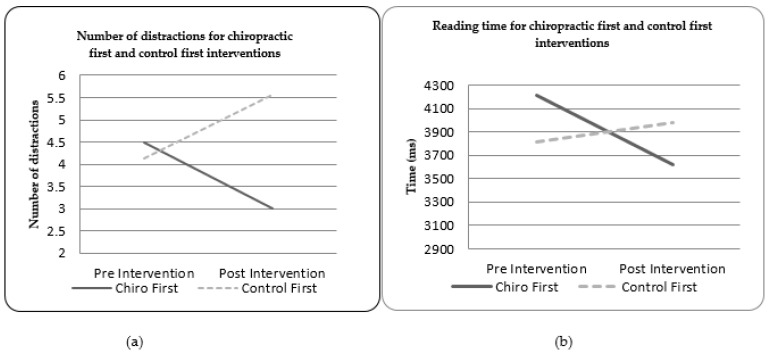
(**a**) Number of target distractions for the chiropractic first and control first interventions; (**b**) reading time for the chiropractic first and control first interventions.

**Figure 2 brainsci-11-01047-f002:**
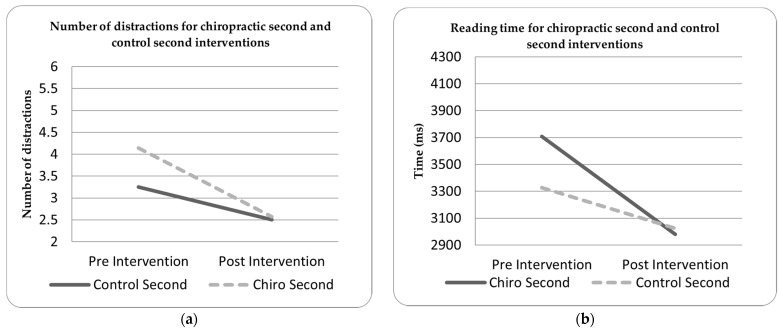
(**a**) Number of target distractions for the chiropractic second and control second interventions; (**b**) reading time for the chiropractic second and control second interventions.

**Table 1 brainsci-11-01047-t001:** Sample Characteristics.

Sample Characteristics	Total *n* = 30	Control First *n* = 15	Chiropractic First *n* = 15	*p* Value
Mean (SD) Age (Years) at assessment	11.3 (2.29)	11.4 (2.23)	11.1 (2.42)	0.72
Grouped age, *n* (%)				
8–11 yrs	17 (56.67)	9 (60.00)	9 (60.00)	
12–15 yrs	13 (43.33)	6 (40.00)	7 (46.67)	
Gender, *n* (%)				0.72
Female	9 (30.00)	5 (33.33)	4 (26.67)	
Male	21 (70.00)	10 (66.67)	11 (73.33)	
Medication Use ***, *n* (%)				0.08
Yes	16 (53.33)	8 (53.33)	8 (53.33)	
No	9 (30.00)	6 (40.00)	3 (20.00)	
Not taken on day of data collection	5 (16.67)	1 (6.67)	4 (26.67)	
Prior Chiropractic Care, *n* (%)				0.04 *
Yes	12 (40.00)	8 (53.33)	4 (26.67)	
No	18 (60.00)	7 (46.67)	11 (73.33)	
Co-morbid Conditions, *n* (%)	16 (53.33)	10 (66.67)	6 (40.00)	0.04 *
ASD	3 (10.00)	2 (13.33)	1 (6.67)	
Other Behavioural Disorders (NOS **)	1 (3.33)	1 (6.67)	0 (0.00)	
Dyspraxia	1 (3.33)	0 (0.00)	1 (6.67)	
Dyslexia	5 (16.67)	4 (26.67)	1 (6.67)	
Other Learning Difficulties (NOS **)	6 (20.00)	3 (20.00)	3 (20.00)	

Note: Data are presented as *n* (%), unless stated otherwise. * Significant *p* < 0.05, ** NOS = Not Otherwise Specified, *** Medication use for the treatment of ADHD.

**Table 2 brainsci-11-01047-t002:** Target task outcome measure results of the study participant’s pre- and post-chiropractic intervention versus active control.

	Intervention								Significance	
	Chiropractic				Control				Group Effect *p* Value	Group by Time Interaction *p* Value
Outcome	*n*	Pre	*n*	Post	*n*	Pre	*n*	Post		
Visual target acquisition time (ms) M (SD)	25 ^†^	1550.00 (3583.84)	24 ^†^	906.63 (1502.63)	25 ^†^	981.95 (1437.58)	24 ^†^	983.29 (2047.83)	0.23	0.23
Number of distractions M (SD)	10 *	4.20 (2.76)	8 *	2.73 (1.68)	9 *	3.82 (2.75)	7 *	4.46 (3.01)	^	^

Note: *n*
^†^ are participants able to calibrate, *n* * are participants who exhibited a distractor behaviour. *p* value significance set at 0.05. ^ = unable to perform group effect or group by time analysis as data was not normally distributed even after attempted data transformation to a normal distribution.

**Table 3 brainsci-11-01047-t003:** Reading task: Outcome measure results of study participant’s pre- and post-chiropractic intervention versus active control.

		Intervention								Significance	
		Chiropractic Intervention				Control Intervention				Group Effect *p* Value	Group by Time Interaction *p* Value
Outcome +		*n*	Pre	*n*	Post	*n*	Pre	*n*	Post		
Reading time	M (SD)	25 ^†^	4016.93 (1894.50)	23 ^†^	3370.06 (1508.91)	23 ^†^	3548.76 (1637.85)	24 ^†^	3440.41 (1971.01)	0.034	0.035
Number of forward fixations	M (SD)	25 ^†^	2.37 (1.14)	23 ^†^	2.13 (0.96)	23 ^†^	2.24 (1.15)	24 ^†^	2.18 (1.00)	^	^
Forward fixation time	M (SD)	25 ^†^	970.94 (645.49)	23 ^†^	894.16 (514.45)	23 ^†^	854.42 (469.52)	24 ^†^	885.23 (604.58)	^	^
Number reverse fixations	M (SD)	18 *	1.21 (0.45)	20 *	1.04 (0.21)	16 *	1.22 (0.46)	19 *	1.17 (0.59)	^	^
Reverse fixation time	M (SD)	18 *	867.15 (672.2)	20 *	755.39 (555.85)	16 *	678.89 (506.73)	19 *	727.23 (559.7)	^	^
Forward saccade length	M (SD)	25 ^†^	407.61 (135.41)	23 ^†^	400.86 (117.50)	23 ^†^	398.41 (96.74)	24 ^†^	392.77 (88.19)	^	^
Reverse saccade length	M (SD)	18 *	284.24 (280.78)	20 *	223.12 (150.09)	16 *	253.37 (191.97)	19 *	223.76 (142.95)	^	^

Note: + = per sentence. *n*
^†^ are participants able to calibrate, *n* * are participants who exhibited a reverse reading behaviour. Time is milliseconds and length in pixels. *p* value significance set at 0.05. ^ = unable to perform group effect or group by time analysis as data was not normally distributed after attempted data transformation to a normal distribution.

## Data Availability

The data presented in this study are available in Appendix A.

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
