# Peer review of "The Effects of Spinal Manipulation on Oculomotor Control in Children with Attention Deficit Hyperactivity Disorder: A Pilot and Feasibility Study†"

_brainsci, 2021, doi:10.3390/brainsci11081047_

Round 1

Reviewer 1 Report

There are major flaws in the methods. I must recommend rejection of this paper.

  • Methods:
    • The measurement tool: “This type of task has been used…” This description is too vague for a measurement used in an investigation. Either the literature provides evidence supporting the exact task used in the current investigation or it does not. The references provided did not support the validity or reliability of the measurement tool, the Eye Tribe Tracker. This is a fatal flaw in my opinion. It also brings into question the use of references by the authors. I recommend checking all references.
      • Reference 27 is the Tribe Eye Tracking website (https://www.theeyetribe.com/devs). I tried to visit it but received a warning that the site posed a security risk to my computer.
      • Reference 11 refers to a 2012 investigation that did not use the Eye Tribe Tracker.
      • Reference 28 leaves out co-authors of the published study in Psychology, and may refer to the first author’s PhD thesis at Indiana State University, 2008. Neither the article in Psychology nor the thesis mentions the Eye Tribe Tracker and in fact the author used different equipment.
    • Why was the diagnosis of ADHD allowed as “parent reported”? Why was ADHD testing not done or doctor’s diagnoses not sought? This brings into question the representative legitimacy of the sample under investigation and therefore the entire study.
    • 53% of participants had co-morbidities, including dyslexia, dyspraxia, and ASD, yet these were not considered excludable as “seriously affecting reading”? It seems to me that these conditions all do seriously affect reading, so why were these participants not excluded?
    • This is a pilot study but there was no discussion of sample size needs (i.e. power calculation) for the full study to follow. Without a power calculation, a pilot study can only investigate methods feasibility, and not draw conclusions on effects of the piloted intervention. Yet, positive conclusions were stated about the potential effectiveness of the intervention.
      • Sim, J. and Lewis, M. (2012) ‘The size of a pilot study for a clinical trial should be calculated in relation to considerations of precision and efficiency’, Journal of clinical epidemiology, 65(3), pp. 301–308. doi: 10.1016/j.jclinepi.2011.07.011.
      • In, J. (2017) ‘Introduction of a pilot study’, Korean journal of anesthesiology, 70(6), pp. 601–605. doi: 10.4097/kjae.2017.70.6.601.
      • Kenis, S. . et al. (2020) ‘Do radiological research articles apply the term “pilot study” correctly? Systematic review’, Clinical radiology, 75(5), pp. 395.e1–395.e5. doi: 10.1016/j.crad.2019.11.010.
    • Another reference issue:
      • “Oculomotor difficulties adversely affect children’s capacity for cognitive processing of visual information and can make simple reading tasks challenging [14].” This reference from 2005 is primarily about epilepsy and secondarily about the association of ADHD with epilepsy. The words ocular, oculomotor, and eye do not appear in it.

After finding these problems I stopped reviewing the paper formally, but there were additional issues with reporting and conclusions.

Author Response

Dear Reviewer 1, 

Thank you for your time and effort in reviewing this manuscript. I have responded to your comments point by point below.

Many thanks,

Alice Cade

Reviewer 1:

  1. Statement of “This type of task has been used…”: The statement “This type of task has been used…” has been amended to read “Reading and target acquisition tasks have previously been used….”
  2. The https://www.theeyetribe.com/devs website: This site’s security certificate expired about a year ago, according to the online warning. I am afraid I am unable to correct this issue at this time as it is out of my control. This is perhaps because the Eye Tribe company has been bought out by the Oculus eye tracking company and its patent company of Facebook (https://venturebeat.com/2016/12/28/facebook-acquires-eye-tracking-company-the-eye-tribe/).
  3. References 11 & 28 Referencing studies that do not use the Eye Tribe Tracker: A study by Ooms et al (2015) that compared low-cost eye trackers, including the eye tribe tracker, to high cost eye trackers (SMI RED250 or Tobii T120 which are systems comparable to the tracker used in reference 28) reported that low-cost eye trackers accuracy and precision were comparable to higher cost trackers (1). Reference 11 does not report the type of eye tracker it used.
  4. Reference 28 missing author names: This has ben corrected within the manuscript.
  5. Parent reported diagnosis of ADHD: As this study was a pilot study preparing for a potential further study the decision was made to use parent reported diagnosis of ADHD however, a larger study may be wise to follow these suggestions and require formal evidence of an ADHD diagnosis.
  6. Comorbidities:53% of participants did have comorbidities such as dyslexia, dyspraxia and ASD. The reason these comorbid disorders were not used as exclusion criteria for this study is that there is still controversy surrounding the identification and differentiation of reading difficulties and oculomotor signs and symptoms all of these disorders (2,3). Due to this issue children with ADHD who also had certain comorbid disorder, such as the ones above were still considered eligible for this study as there is a large overlap in symptomatology and presentation.
  7. Full study sample size needs: If a full study were to follow this one it would include power calculations based on this study. At this stage a following study has not been planned. However, I have included sample power calculations in the manuscript to reflect the sample size needed if a follow-up study were to be undertaken.
  8. Reference 14: The Dunn and Kronenberger reference has been updated to a potentially more acceptable Dunn and Kronenberger (4)

References

  1. Ooms K, Krassanakis V. Measuring the spatial noise of a low-cost eye tracker to enhance fixation detection. J Imaging [Internet]. 2018 [cited 2021 Mar 9];4(8). Available from: www.mdpi.com/journal/jimaging
  2. Bilbao C, Piñero DP. Diagnosis of oculomotor anomalies in children with learning disorders. Clin Exp Optom [Internet]. 2020 Sep 1 [cited 2021 Jul 23];103(5):597–609. Available from: https://onlinelibrary.wiley.com/doi/full/10.1111/cxo.13024
  3. Åsberg J, Dahlgren SO, Dahlgren Sandberg A. Basic reading skills in high-functioning Swedish children with autism spectrum disorders or attention disorder. Res Autism Spectr Disord. 2008 Jan 1;2(1):95–109.
  4. Dunn DW, Kronenberger WG. Attention deficit. Handb Clin Neurol. 2013/04/30. 2013;111:257–61.

Reviewer 2 Report

I have no suggestions for improvement. I would like to congratulate the authors on an extremely high-quality study!  It serves as an excellent example of how a pilot/feasibility study should be conducted!  There would be more high-quality RCTs if more investigators conducted such thorough, impeccably reasoned preliminary studies.  

Author Response

Dear Reviewer,

Thank you for your review, I very much appreciate the time you took in reviewing this submission.

Reviewer 3 Report

This was an interesting study, and I applaud the authors' effort to submit chiropractic to scientific scrutiny.   I am recommending publications, but I would like to see some strong statement that sets forth the limits of the study.  This strikes me as well done preliminary research, but I wouldn't want a reader to mistakenly over interpret  it. 

My primary concern is the sample size.   27 participants is good for a pilot study, but more definitive work would require a substantially larger sample.  This is really why I would like a statement regarding the limits of the study,

I appreciate the authors' observation that perhaps the cross over design was affected by cross over effects.  Perhaps in subsequent research (and I hope the authors continue this line of investigation), they might consider either a multiple baseline design or an institutional cycle design.  They would eliminate the cross over effects. 

Medications - authors mention use of medications by the participants.   It would have been interesting to know which medications.  Specifically, were all of the medications prescribed for ADHD, or were other medications involved.   

Table 2 - Because most parents didn't complete these questions, I'd suggest deleting the table entirely.    

Author Response

Dear Reviewer,

Thank you for your review, I very much appreciate the time you took in reviewing this submission. I have included a clearer statement regarding limitations based on your advice and have removed table 2.

I have also updated the manuscript to reflect that all the medications used were for the treatment of ADHD, but in any future studies care will be taken to collect and describe the type of medication used.

Thank you again for your time and effort.

Round 2

Reviewer 1 Report

Brain Sciences review R1

  1. Regarding the authors’ failure to provide validity/reliability data on the measurement tool, the reference they now offer has not been inserted into the revised manuscript. In addition, the reference cited notes specific limitations to the use of the Eye Tribe Tracker (e.g. “high-accuracy saccades”). Any limitations relating to the equipment must be discussed in detail for results to be meaningfully interpreted. The newly cited paper (Ooms) also cites two other papers that evaluated the Eye Tribe Tracker, a different one by Ooms and one by Dalmaijer (2014), yet these other two papers are not cited. Do they support the authors’ use of this tool? This is representative of the referencing problems throughout the paper.
    1. In addition, Dalmaijer noted that ‘[He] evaluated the question “Is the low-cost EyeTribe eye tracker any good for research?”. He concluded that the Eye Tribe Tracker can be appropriate for scientific research related to fixation analyses and pupillometry, but not for evaluating high accuracy saccadic metrics. This latter issue is related to the low sampling rate of 60 Hz.’ Saccades were a metric in the current paper. How did this shortcoming affect the results?
  2. The issue of lack of calibration of the equipment was not addressed (fatal flaw).
  3. The issue of comorbidities and exclusion from the study was not adequately addressed. In particular, dyslexia is specifically a reading disorder. A much more specific rationale for including these participants must be given for each of the disorders included.
  4. The authors did not address the issue of checking all references to ensure that they were accurate and appropriate. Given the new reference issues in the revision, this indicates a continued lack of attention to the problem.
  5. The positive conclusions in the Discussion remain, despite the fact that this pilot study cannot make any conclusions due to the many flaws in methodology and lack of adequate sample size:
    1. Nearly all of the reading and target outcomes improved post spinal manipulation indicating better reading ability and decreased distractibility [11,44]. The caveat of this discussion is that, excepting reading time, none of the data were statistically significant, thus it can only be said to show trends or possibilities in the data that warrant further investigation.’ This is untrue.
    2. ‘Nevertheless, findings suggest that adjusting the spine may be associated with improvements in oculomotor control relating to children’s reading times.’ This is untrue.
  6. In summary, the critical flaws indicated in my previous review have not been addressed (indeed, it is unlikely that the methodological ones even could be addressed at this stage) and I must again recommend rejection.